# Fasting plasma glucose levels are associated with all-cause and cancer mortality: A population-based retrospective cohort study

**Qing Gao**[1]⊙, **Qi Wang**[1]⊙, **Zhijuan Gan**[2], **Meng Wang**[3], **Dafeng Lu**[2], **Bingdong Zhan**[1,2]*

1 School of Public Health, Zhejiang Chinese Medical University, Hangzhou, Zhejiang Province, China, 2 Quzhou Center for Disease Control and Prevention, Quzhou, Zhejiang Province, China, 3 Zhejiang Provincial Center for Disease Control and Prevention, Hangzhou, Zhejiang Province, China

⊙ These authors contributed equally to this work.
* bd_zhan@126.com

**Data Availability Statement:** Data cannot be shared publicly because of legal and ethical considerations. Data are available from the Ethics Review Committee of Quzhou Center for Disease

## Abstract

Despite a growing body of research indicating a link between fasting glucose levels and mortality, the relationship between fasting glucose and all-cause and cancer mortality remains inconsistent. In this study, we used Cox regression and restricted cubic spline models to analyze the association and dose-response relationship between fasting plasma glucose levels and all-cause and cancer mortality in a retrospective cohort based on data from the 2015 health check-ups of residents in Quzhou City. After a mean follow-up of 5.31 years for 148,755 study participants, 10,345 deaths occurred, with an all-cause mortality density of 131.09/10,000 person-years, of which 2,845 were cancer deaths, with a cancer mortality density of 36.05/10,000 person-years. There was a "J" shaped dose-response relationship between fasting plasma glucose levels and all-cause and cancer mortality. Relative to normal fasting glucose (NFG), the risk of all-cause mortality (HRs and 95% CIs) in the impaired fasting glucose (IFG) and diabetes mellitus (DM) groups was 1.11 (1.06, 1.16) and 1.43 (1.35, 1.52), respectively, and the risk of cancer mortality in the DM group was 1.22 (1.09, 1.37). In this cohort study, we found that fasting plasma glucose levels were significantly associated with the risk of all-cause and cancer mortality.

## Introduction

In recent years, with socioeconomic development and the improvement of people's living standards, the prevalence of diabetes worldwide has been increasing. According to the latest report of the International Diabetes Federation (IDF), about 537 million adults worldwide have diabetes in 2021; the number is expected to rise to 643 million by 2030; and to 783 million by 2045 [1]. China is the largest developing country in the world. It is reported that 114 million adults in China suffer from diabetes, the prevalence rate is 11.6% [2]. Diabetes has become one of the serious public health problems in China [3]. Diabetes is an important risk factor for death, placing a huge burden on countries around the world [4, 5].

Control and Prevention (contact via 493440928@qq.com) for researchers who meet the criteria for access to confidential data.

**Funding:** The author(s) received no specific funding for this work.

**Competing interests:** The authors have declared that no competing interests exist.

Diabetes can lead to cardiovascular, gastrointestinal, immune and nervous system disorders, can also lead to complications such as kidney failure, heart problems, vision loss, disability, high mortality rate [6]. Diabetes is estimated to cause 4.2 million deaths among adults aged 20–79 years in 2019, accounting for 11.3% of global mortality [7]. According to the National Bureau of Statistics, the number one cause of death in China between 2000 and 2020 has been cancer [8]. Global Cancer Data 2020 (GLOBOCAN 2020) shows [9] that cancer deaths in China account for 30 percent of global cancer deaths. Some studies suggest that high fasting plasma glucose (FPG) is an important risk factor for cancer death [10, 11]. A prospective cohort study on the relationship between diabetic patients and mortality in Korea concluded that FPG showed a U-shaped relationship with all-cause mortality, and that hypoglycemia increased mortality in patients with DM [12]. Findings from the Eastern Mediterranean Region (EMR) and Taiwan, China support this conclusion [13, 14], but some studies have also found a higher risk of death in the high FPG group [15, 16]. Although a growing body of research has uncovered links between FPG and all-cause and cancer mortality, the two diseases are not globally consistent in their etiological data.

In this study, based on data from health checkups of residents in Quzhou City from 1 January 2015 to 31 December 2015, we used a retrospective cohort study to analyze the effects of different glycemic status on all-cause and cancer mortality, and to further explore the dose-response relationship between FPG levels and outcomes.

## Methods

### Study design and population

This study retrospectively collected the health examination information of residents in Quzhou City, Zhejiang Province from January 1, 2015 to December 31, 2015. Each subject was examined at baseline by a local village doctor, who collected all subjects' health examination information by questionnaire, physical examination, laboratory examination and auxiliary examination. Through professional research technicians in the health examination information to collect the characteristics of each subject data. All information collected is quality controlled by trained research technicians to form a database of information on health check-ups of the population. The study accessed the database that removes identifiable individual participant information on November 20, 2023 to obtain the data, using the registration number as the only identifiable information. The use of this database was reviewed and approved by the Ethics Review Committee of Quzhou Center for Disease Control and Prevention (CDC), IRB No. 2023-023-01. All participant identifying information was anonymized and therefore exempt from informed consent.

A uniform methodology was used to conduct face-to-face interviews with the study participants, and basic information was entered via a tablet computer; basic demographic characteristics obtained included age and gender; and lifestyle information included physical activity, alcohol consumption, and smoking. Smoking was categorized as never smoking, quit smoking and smoking; smoking was defined as having smoked at least 100 cigarettes or having smoked in the month before the survey or currently smoking; otherwise, it was defined as never smoking; former smokers who had quit smoking for more than six months were defined as having quit smoking. Alcohol consumption was categorized as never drinking, occasional drinking, regular drinking and daily drinking; occasional drinking was defined as less than or equal to one drink per week and regular drinking was defined as more than one drink per week (excluding one drink) but less than daily drinking. Physical activity was categorized as daily, more than once a week, occasionally and never. Exercise was defined as ≥30 minutes per session. The height and weight of all participants were measured using standardized instruments

while wearing light clothing and removing shoes and hats. The body mass index (BMI, kg/m$^2$) was calculated by dividing weight (kg) by the square of height (m). After the subjects had been sitting still for 10 minutes, each participant's seated blood pressure was measured using a sphygmomanometer to obtain information on systolic blood pressure (SBP, mmHg) and diastolic blood pressure (DBP, mmHg), and three measurements were averaged. FPG was measured from venous blood collected from subjects who had fasted for more than 8 hours, and FPG levels were classified into four categories according to the criteria published by the American Diabetes Association [17]: low fasting glucose (LFG, FPG < 3.9 mmol/L), normal fasting glucose (NFG, 3.9 mmol/L ≤ FPG < 5.6 mmol/L), impaired fasting glucose (IFG, 5.6 mmol/L ≤ FPG < 7.0 mmol/L), and diabetes mellitus (DM, FPG ≥ 7.0 mmol/l).

Study subjects with the following characteristics were excluded: (1) subjects younger than 18 years of age; (2) lack of accurate identifying information for follow-up; (3) lack of record of smoking, alcohol consumption, physical activity; (4) continuous variable values are outliers (above or below the mean ± 3 SDs). Finally, a total of 148,755 subjects were included in the analysis. A flowchart of the study is shown in Fig 1.

## Follow-up and endpoint definition

In this study, the subjects were followed up for deaths mainly by linking to the death information system. Death information was obtained by the Quzhou City Center for Disease Control and Prevention on the basis of the death registration cards reported from the monitoring of the streets and townships under its jurisdiction. All death cases were standardized by all levels and types of medical institutions to issue medical certificates of death, complete the network direct reporting, and entered into the death information system after being reviewed by the municipal and county (district) CDCs. The primary endpoints of the study were all-cause and cancer mortality. Causes of death were classified according to the International Classification of Diseases, 10th edition (ICD-10), with all-cause mortality defined as A00-Z99 and cancer mortality defined as C00-C97. Participants were classified from baseline to death or until December 31, 2020, whichever occurred earlier.

## Statistical analysis

Continuous variables were tested for normality; normal information was described by mean ± standard deviation, and comparisons between groups were made by t-test or F-test; non-normal information was described by median (quartiles), and comparisons between groups were made by Kruskal-Wallis H non-parametric test; information on categorical variables was described by frequency (percentage), and comparisons between groups were made by chi-square test. The confounding factors include: sex (categorical variable), age (continuous variable), SBP (continuous variable), DBP (continuous variable), BMI (continuous variable), physical exercise (categorical variable), smoking (categorical variable) and alcohol consumption (categorical variable).

All covariates met the proportional risk assumption based on the Schoenfeld residuals test for trend. Cox proportional risk regression models were used to estimate the association between FPG levels and all-cause and cancer mortality, and the degree of association was assessed by hazard ratio (HR) and 95% confidence interval (95% CI), using the NFG group as the reference group, with model 1 unadjusted for any variable; model 2 adjusted for sex and age; and model 3 adjusted for SBP, DBP, BMI, physical exercise, smoking, and alcohol consumption on the basis of model 2. Subgroup analyzes were performed according to age (< 60 and ≥ 60 years), and sex to explore potential modification effects. We also tested the robustness of the results through sensitivity analyzes, where the study further excluded (1) subjects

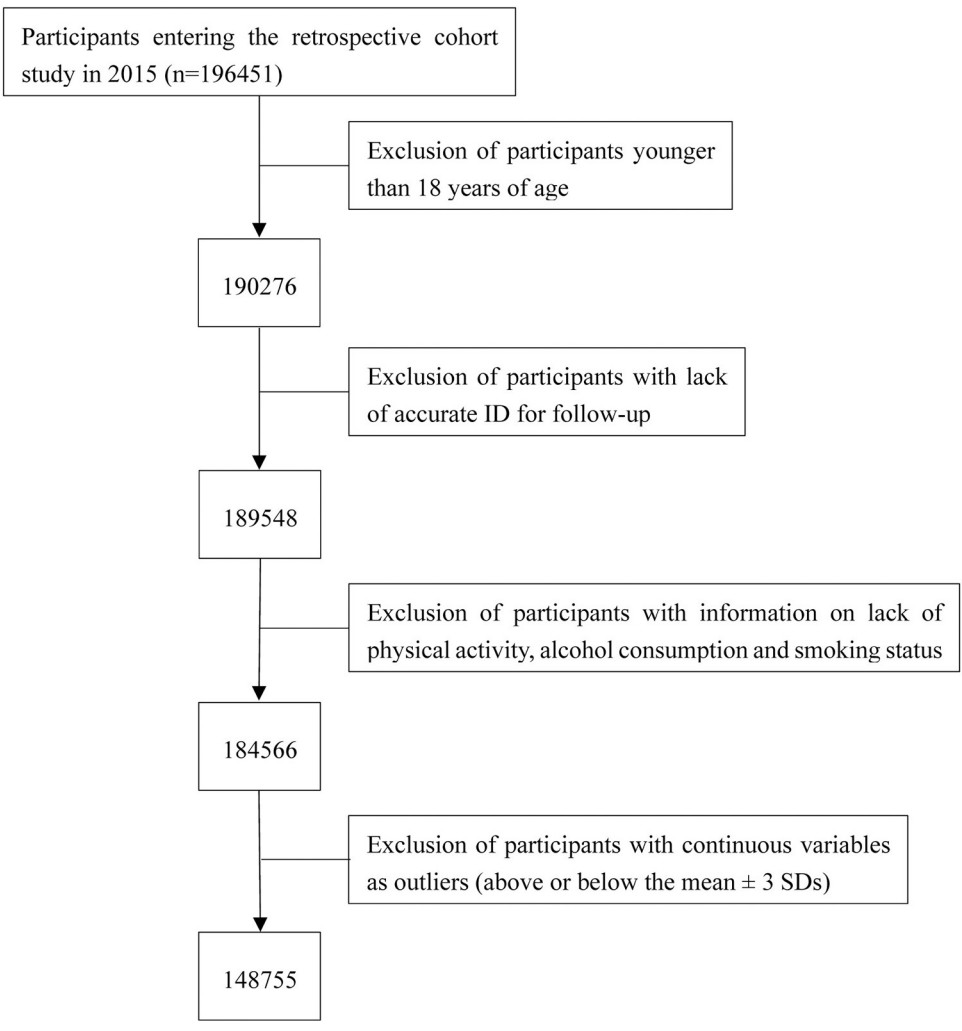

**Fig 1. Flow chart of retrospective cohort study.**

with cancer at baseline, those who died within one year of follow-up, and (2) further excluded baseline smokers from re-analyzes on the basis of (1). The dose-response relationship between FPG and the risk of all-cause and cancer mortality was analyzed using restricted cubic spline models, controlling for confounders, and a non-linear dose-response relationship was considered to exist if $P$-overall $< 0.05$ and a $P$-non-linear $< 0.05$ value. All analyzes were performed using R software (version 4.2.1, R Foundation for Statistical Computing, Vienna, Austria), and a two-sided $P < 0.05$ was considered statistically significant.

## Results

### Baseline characteristics of the participants

A total of 148,755 subjects, including 64,013 men and 84,742 women, with a median age of 60.00 years, were enrolled in this study. Grouped according to FPG, there were 2,520 in the LFG group, 80,456 in the NFG group, 50,414 in the IFG group and 15,365 in the DM group. Differences in the distribution of age, sex ratio, proportion of smokers, proportion of drinkers, physical activity, BMI, SBP, DBP, all-cause mortality and cancer mortality of the study subjects in different FPG groups were statistically significant ($P < 0.05$). Detailed baseline

**Table 1. Baseline characteristics according to different clinical categories of FPG.**

| Characteristic | Overall | LFG | NFG | IFG | DM | P value |
|---|---|---|---|---|---|---|
| No. of cases | 148755 | 2520 | 80456 | 50414 | 15365 | |
| Person-year (total) | 789150.10 | 13406.89 | 428798.23 | 266541.03 | 80403.95 | |
| Sex [n (%)] | | | | | | <0.001 |
| Men | 64013 (43.0) | 1185 (47.0) | 35638 (44.3) | 21088 (41.8) | 6102 (39.7) | |
| Women | 84742 (57.0) | 1335 (53.0) | 44818 (55.7) | 29326 (58.2) | 9263 (60.3) | |
| Age, years [M (P25, P75)] | 60.00 [51.00, 68.00] | 61.00 [51.00, 68.00] | 59.00 [49.00, 67.00] | 62.00 [53.00, 69.00] | 62.00 [55.00, 70.00] | <0.001 |
| SBP, mmHg [M (P25, P75)] | 130.00 [120.00, 140.00] | 128.00 [115.88, 140.00] | 128.00 [120.00, 138.00] | 130.00 [120.00, 140.00] | 134.00 [120.00, 146.00] | <0.001 |
| DBP, mmHg [M (P25, P75)] | 79.00 [70.00, 84.00] | 80.00 [70.00, 86.00] | 78.00 [70.00, 84.00] | 80.00 [72.00, 84.00] | 80.00 [72.00, 86.00] | <0.001 |
| BMI, kg/m$^2$ [M (P25, P75)] | 22.87 [20.96, 24.97] | 22.47 [20.55, 24.44] | 22.68 [20.83, 24.67] | 23.05 [21.10, 25.22] | 23.53 [21.36, 25.81] | <0.001 |
| FPG, mmol/L [M (P25, P75)] | 5.40 [4.93, 6.00] | 3.64 [3.44, 3.80] | 5.00 [4.68, 5.26] | 5.95 [5.70, 6.25] | 8.10 [7.40, 9.60] | <0.001 |
| Cigarette smoking [n (%)] | | | | | | <0.001 |
| Never smoking | 114881 (77.2) | 1763 (70.0) | 61887 (76.9) | 39320 (78.0) | 11911 (77.5) | |
| Former smoking | 6582 (4.4) | 153 (6.1) | 3323 (4.1) | 2296 (4.6) | 810 (5.3) | |
| Current smoking | 27292 (18.3) | 604 (24.0) | 15246 (18.9) | 8798 (17.5) | 2644 (17.2) | |
| Physical exercise [n (%)] | | | | | | <0.001 |
| Every day | 17824 (12.0) | 208 (8.3) | 9877 (12.3) | 6062 (12.0) | 1677 (10.9) | |
| More than once a week | 8328 (5.6) | 94 (3.7) | 4924 (6.1) | 2843 (5.6) | 467 (3.0) | |
| Occasionally | 13777 (9.3) | 186 (7.4) | 7845 (9.8) | 4621 (9.2) | 1125 (7.3) | |
| Never | 108826 (73.2) | 2032 (80.6) | 57810 (71.9) | 36888 (73.2) | 12096 (78.7) | |
| Alcohol consumption [n (%)] | | | | | | <0.001 |
| Never drinking | 114290 (76.8) | 1875 (74.4) | 61944 (77.0) | 38638 (76.6) | 11833 (77.0) | |
| Occasionally drinking | 9800 (6.6) | 171 (6.8) | 5317 (6.6) | 3287 (6.5) | 1025 (6.7) | |
| Often drinking | 7723 (5.2) | 131 (5.2) | 4290 (5.3) | 2617 (5.2) | 685 (4.5) | |
| Every day | 16942 (11.4) | 343 (13.6) | 8905 (11.1) | 5872 (11.6) | 1822 (11.9) | |
| All-cause mortality [n (%)] | 10345 (7.0) | 197 (7.8) | 4725 (5.9) | 3862 (6.7) | 1561 (10.2) | <0.001 |
| Cancer mortality [n (%)] | 2845 (1.9) | 60 (2.4) | 1395 (1.7) | 1021 (2.0) | 369 (2.4) | <0.001 |

Abbreviations: FPG, fasting plasma glucose; LFG, low fasting glucose; NFG, normal fasting glucose; IFG, impaired fasting glucose; DM, diabetes mellitus; BMI, body mass index; SBP, systolic blood pressure; DBP, diastolic blood pressure

characteristics of study participants grouped by different FPG levels are described in Table 1. The all-cause mortality and cancer mortality groups were more likely to be male, older, current smokers, daily drinkers and never physically active, with higher levels of FPG and SBP and lower BMI (S1 and S2 Tables).

## FPG and risk of all-cause mortality

As of 31 December 2020, the cumulative follow-up time was 789,150.10 person-years, with an average follow-up time of 5.31 years, and 10,345 all-cause mortality occurred, with an all-cause mortality density of 131.09/10,000 person-years; of these, 6,017 and 4,328 were male and female patients, respectively, with a mortality density of 179.42/10,000 person-years and 95.37/10,000 person-years. The densities of death for different FPG levels subgroups were 146.94/10,000 person-years, 110.19/10,000 person-years, 144.89/10,000 person-years, and 194.14/10,000 person-years, respectively.

Cox regression analyzes showed that in the total population, with the NFG group as the reference group, in the unadjusted model 1, the corresponding HR values (95% CI) for the LFG, IFG, and DM groups were 1.33 (1.15, 1.53), 1.32 (1.26, 1.37), and 1.77 (1.67, 1.87), respectively. Model 2 showed a decreased risk of all-cause mortality for different FPG levels after adjusting

for sex and age, but still showed a positive association (HR of 1.21, 1.08, 1.40). After adjusting for baseline SBP, DBP, BMI, physical exercise, smoking, and alcohol consumption based on model 2, the risk of death increased by 11% and 43%, respectively, in the IFG and DM groups compared with the NFG group, and the HR values and 95% CIs were 1.11 (1.06, 1.16) and 1.43 (1.35, 1.52). The risk of death in the LFG group was not statistically different from the NFG group (HR of 1.14, 0.99, 1.31). (Table 2).

Restricted cubic spline results showed a significant J-shaped dose-response relationship for the association between FPG and the risk of all-cause mortality in the total population after adjusting for relevant confounders (*P*-overall < 0.001, *P*-non-linear < 0.001). The FPG concentration associated with the lowest risk of all-cause mortality was 4.99 mmol/L (HR of 0.97, 0.96, 0.99), and the risk of all-cause mortality was increased by either too high or too low a concentration of FPG, with HR decreasing with increasing FPG when FPG was less than 3.82 mmol/L, and increasing with FPG when FPG was greater than 5.51 mmol/L (Fig 2).

## FPG and risk of cancer mortality

During the follow-up period, a total of 2,845 study subjects died from cancer, with a cancer mortality density of 36.05/10,000 person-years, of which 1,859 were males and 986 were females, with mortality densities of 55.43/10,000 person-years and 21.73/10,000 person-years. The results of Cox regression analyzes showed that in the total population, in the unadjusted model 1, the LFG, IFG and DM groups showed an association with the risk of cancer mortality relative to the NFG group, corresponding to HR values (95% CI) of 1.37 (1.06, 1.78), 1.18 (1.09, 1.28), and 1.41 (1.26, 1.59), and that, in Models 2 and 3, only the DM group showed a statistically significant difference in the risk of cancer mortality relative to the NFG group, with HR and 95% CI of 1.22 (1.09, 1.37) (Table 3).

Restricted cubic spline results showed a significant J-shaped dose-response relationship (*P*-overall < 0.001, *P*-non-linear = 0.047) for the association between FPG and risk of cancer mortality in the total population, adjusting for relevant confounders, with the concentration of FPG associated with the lowest risk of cancer mortality being 5.12 mmol/L (HR of 0.99, 0.97, 1.01), and HR increased with increasing FPG when FPG was greater than 6.29 mmol/L (Fig 3).

## Subgroup analyzes of the association between the FPG and risk of mortality

In the analysis of all-cause mortality, stratified by gender, both men and women in the IFG and DM groups showed an increased risk of all-cause mortality compared to the NFG group

**Table 2. HR (95% CI) for risk of all-cause mortality by fasting glucose category.**

| Model HR (95%CI) | Fasting Glucose Categories | | | |
|---|---|---|---|---|
| | LFG | NFG | IFG | DM |
| Death density (per 10,000 person-years) | 146.94 | 110.19 | 144.89 | 194.14 |
| Model 1 | 1.33 (1.15,1.53) ** | 1.00 (REF) | 1.32 (1.26,1.37) ** | 1.77 (1.67,1.87) ** |
| Model 2 | 1.21 (1.05,1.39) ** | 1.00 (REF) | 1.08 (1.04,1.13) ** | 1.40 (1.33,1.49) ** |
| Model 3 | 1.14 (0.99,1.31) | 1.00 (REF) | 1.11 (1.06,1.16) ** | 1.43 (1.35,1.52) ** |

*P<0.05

**P<0.001

Model 1: unadjusted

Model 2: adjusted for age, sex

Model 3: adjusted for SBP, DBP, BMI, physical exercise, smoking, and alcohol consumption on the basis of the Model 2

Abbreviations: LFG, low fasting glucose; NFG, normal fasting glucose; IFG, impaired fasting glucose; DM, diabetes mellitus

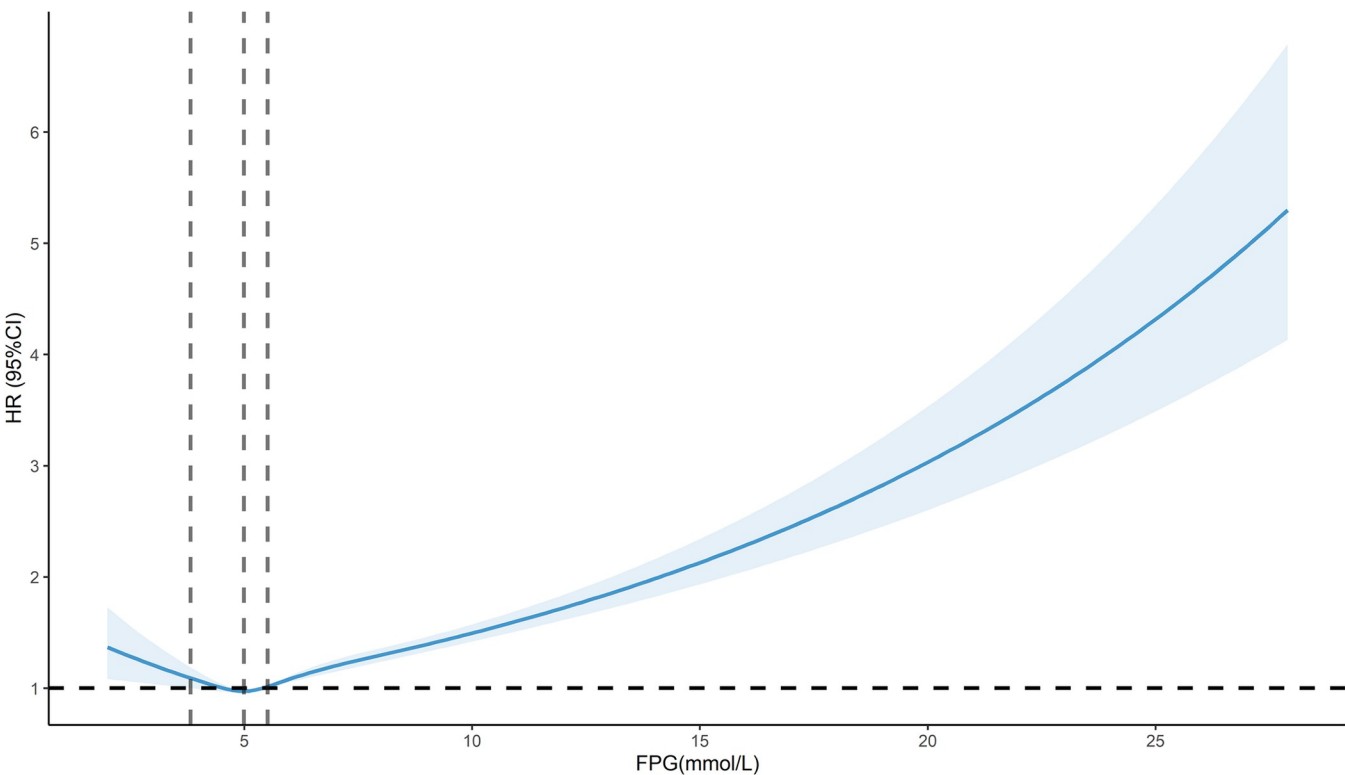

**Fig 2. Dose-response relationship between FPG and risk of all-cause mortality.**

in Model 3, after adjusting for all confounding factors. In the age-stratified analysis, among individuals younger than 60 years, Model 3 results showed an increased risk of all-cause mortality in both the IFG and DM groups compared to the NFG group, with HR and 95% CI of 1.36 (1.18, 1.57) and 2.41 (2.01, 2.88), respectively. Among individuals aged 60 years and older, the IFG and DM groups also had an increased risk of all-cause mortality compared to the NFG group, with HR and 95% CI of 1.21 (1.15, 1.26) and 1.53 (1.44, 1.63), respectively.

In the analysis of cancer mortality, stratified by gender, both men and women in the DM group showed an increased risk of cancer mortality compared to the NFG group in Model 3.

**Table 3. HR (95% CI) for risk of cancer mortality by fasting glucose category.**

| Model HR (95%CI) | Fasting Glucose Categories | | | |
|---|---|---|---|---|
| | LFG | NFG | IFG | DM |
| Death density (per 10,000 person-years) | 44.75 | 32.53 | 38.31 | 45.89 |
| Model 1 | 1.37 (1.06,1.78) * | 1.00 (REF) | 1.18 (1.09,1.28) ** | 1.41 (1.26,1.59) ** |
| Model 2 | 1.24 (0.96,1.61) | 1.00 (REF) | 1.03 (0.95,1.12) | 1.20 (1.07,1.35) * |
| Model 3 | 1.19 (0.92,1.54) | 1.00 (REF) | 1.05 (0.97,1.14) | 1.22 (1.09,1.37) ** |

*$P<0.05$

**$P<0.001$

Model 1: unadjusted

Model 2: adjusted for age, sex

Model 3: adjusted for SBP, DBP, BMI, physical exercise, smoking, and alcohol consumption on the basis of the Model 2

Abbreviations: LFG, low fasting glucose; NFG, normal fasting glucose; IFG, impaired fasting glucose; DM, diabetes mellitus

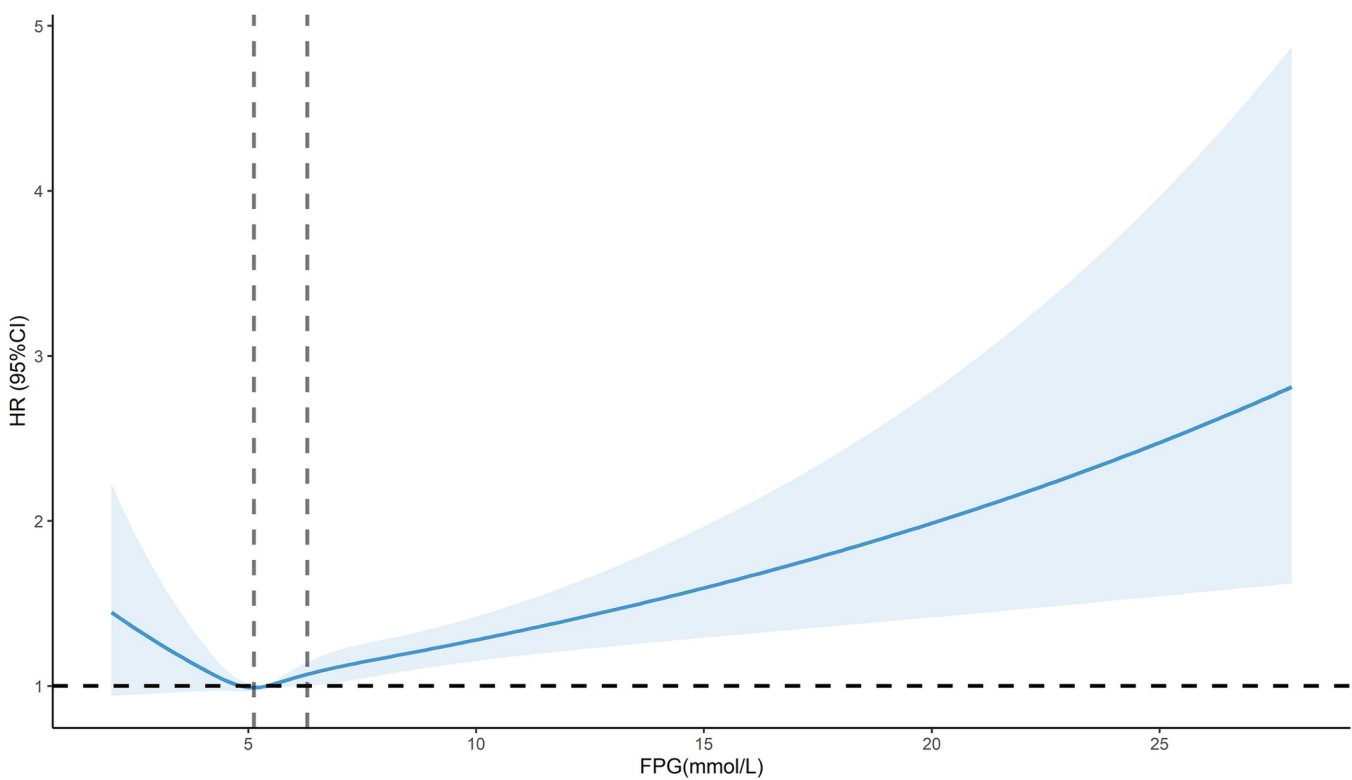

**Fig 3. Dose-response relationship between FPG and risk of cancer mortality.**

In the age-stratified analysis, among individuals younger than 60 years, the IFG and DM groups showed an increased risk of cancer mortality compared to the NFG group, with HR and 95% CI of 1.31 (1.06, 1.63) and 1.84 (1.37, 2.48), respectively. Among individuals aged 60 years and older, the DM group showed an increased risk of cancer mortality compared to the NFG group (HR 1.25, 95% CI 1.10, 1.41) (S3 Table).

## Sensitivity analysis

To verify the stability of the findings, further sensitivity analyzes were performed excluding those with cancer at baseline (n = 1,910), those who died within one year of follow-up (n = 1,082), and those who smoked at baseline (n = 27,292), and the results showed that there was no significant change in the risk of all-cause mortality for any of the different FPG levels, and that excessively high FPG was still associated with an increased risk for the occurrence of all-cause mortality, and that there was stability in the results, as shown in S4 Table. In the Cancer Causes of Death study, the risk of cancer mortality was consistent with the original study when only those who had cancer at baseline and died within one year of follow-up were excluded; however, when the baseline smokers were further excluded, the association of the DM group with cancer mortality was not statistically significant after adjusting for potential confounders, as shown in S5 Table.

## Discussion

In this study, we discussed the association between FPG and all-cause mortality and cancer mortality using a retrospective cohort in a Chinese population with a total of 10,345 deaths

among 148,755 subjects, of which 2,845 were cancer mortality, during a median follow-up of 5.31 years. We found a higher risk of all-cause mortality in the IFG group compared with the NFG group, and a higher risk of all-cause mortality and cancer mortality in the DM group. There was also a J-shaped dose-response relationship between FPG levels and all-cause and cancer mortality.

## DM and all-cause and cancer mortality

National and international studies have concluded that patients with abnormal glucose metabolism have a higher incidence of malignancy and mortality than healthy groups [18, 19]. We consider diabetes to be an important risk factor, and a meta-analysis including 35 studies found a 30% increased risk of all-cause mortality in patients with diabetes [20]. An analysis of three cohort studies from the United States found that the risk of all-cause and cancer mortality in diabetic populations was up to 79% higher than normal in both men and women [21]. The analysis may be that on the one hand diabetes can lead to multiple complications such as chronic kidney disease, plaque formation and atherosclerosis [22]. On the other hand, diabetes leads to the long-term accumulation of advanced glycosylation end products, which can increase intracellular inflammatory factors and oxidative stress, leading to DNA damage and cancerous transformation of cells [23]. In addition, cancer cell growth and proliferation depend on glycolysis for energy, and hyperglycaemia can provide an adequate source of glucose for rapidly proliferating cancer cells [24].

## IFG and all-cause and cancer mortality

Our study showed a strong association between IFG and all-cause mortality, but the association was not significant for cancer mortality. In a meta-analysis of prediabetes and risk of cancer mortality, IFG was found to be associated with only a few site-specific cancers, such as liver cancer [25]. An Australian prospective study (Australian Diabetes, Obesity, and Lifestyle Study, AusDiab) found that after 5.2 years of follow-up, IFG increased the risk of all-cause mortality by 60%, the risk of CVD mortality by 150%, and made no contribution to cancer mortality [26]. A study in Taiwan, China, found similar results [27]. It has therefore been suggested that the increased risk of all-cause mortality in prediabetes may be due to microvascular events [28].

## LFG and all-cause and cancer mortality

All-cause mortality has been shown to be significantly increased in patients with LFG [29, 30]. A joint study between China and the United States showed that the correlation between LFG and cancer and other causes of death was not significant [31], and a similar situation was confirmed in a prospective study in Sweden [32], which is consistent with the results of our study, and FPG was not associated with all-cause and cancer mortality in our study. However, the relationship between LFG and mortality is unclear and the results vary widely, so the association between LFG and mortality risk remains to be verified.

## FPG levels and all-cause and cancer mortality

We found a J-shaped association between FPG levels and the risk of all-cause and cancer mortality in our study population, with the lowest risk of all-cause mortality at a FPG level of 4.99 mmol/L, and both low and high FPG levels associated with an increased risk of all-cause mortality. A prospective cohort study from Korea found the lowest risk of mortality with FPG levels of 4.44–5.22 mmol/L [33]. Another Korean study found a U-shaped association between

FPG and all-cause mortality [12]. The FPG levels associated with the lowest risk of cancer mortality was 5.12 mmol/L, and high FPG levels were associated with an increased risk of cancer mortality. A Japanese study found a dose-response relationship between FPG and pancreatic cancer mortality [34]. A J-shaped relationship between FPG and cancer mortality was also found in a cohort study conducted in rural China [35].

### FPG and all-cause and cancer mortality in participants with different characteristics

In the subgroup analysis by gender and age, it was found that the risk of all-cause mortality in the IFG and DM groups and the risk of cancer mortality in the DM group increased to varying degrees across different genders, with women having a slightly higher mortality risk than men. The analysis of gender differences in FPG and all-cause mortality may be due to intrinsic biological differences or differences in the management of risk factors. For example, specific changes in sex hormones, visceral fat, and muscle mass between the sexes can modulate glucose metabolism [36, 37]. IFG and DM increase the risk of all-cause mortality, and DM increases the risk of cancer mortality, with this association being more pronounced in individuals under 60 years of age. A study by Yi et al. in Korea also found similar results [33], showing that within the prediabetes range, each increase in fasting blood glucose level in young adults was generally associated with a stronger HR and a relatively stronger correlation with increased mortality compared to older adults.

Sensitivity analyzes after excluding those with cancer at baseline who died within one year of follow-up showed that the study results were consistent with the total population; after further exclusion of those who smoked at baseline, the all-cause risk of death was consistent with the total population, while the risk of cancer mortality turned out to be unassociated, so that smoking had an impact on the results. It has been shown that smoking plays a leading role in cancer-related deaths caused by modifiable risk factors in China, contributing to 23% to 26% of cancer-related deaths [38, 39].

This study has a number of strengths: it was a population-based retrospective cohort study, the sample size was large, including 148,755 subjects, the mortality data were reviewed by municipal and county (district) CDC, and, in addition, a restricted cubic spline methodology was used, which allowed for the intuitive reflection of the dose-response relationship between FPG levels and outcomes.

There are also some limitations of this study that are worth noting; firstly, as the study data were obtained from a population in the Quzhou area of China, our conclusions are only applicable to this population and caution is needed when extrapolating the results; secondly, our follow-up time was relatively insufficient, with a median follow-up time of only 5.31 years; and lastly, the present study only focused on the relationship between baseline FPG levels and death, and did not evaluate the changes in FPG and the mortality risk, which needs to be further explored in future studies.

### Conclusions

In conclusion, the present study demonstrated that the IFG group significantly increased the risk of all-cause mortality relative to the NFG group, and that the DM group significantly increased the risk of all-cause and cancer mortality. In addition, a J-shaped dose-response relationship was observed between FPG levels and all-cause and cancer mortality. Therefore, early monitoring and intervention strategies should be introduced in the population with abnormal glucose metabolism to reduce the risk of death.

## Supporting information

**S1 Table. Basic demographic characteristics of the all-cause mortality group.**
(DOCX)

**S2 Table. Basic demographic characteristics of the cancer mortality group.**
(DOCX)

**S3 Table. Subgroup analyzes of the association between the FPG and risk of mortality.**
(DOCX)

**S4 Table. Sensitivity analysis of different FPG levels associated with all-cause mortality.**
(DOCX)

**S5 Table. Sensitivity analysis of the association of different FPG levels with cancer mortality.**
(DOCX)

## Author Contributions

**Conceptualization:** Bingdong Zhan.

**Data curation:** Qi Wang, Zhijuan Gan.

**Formal analysis:** Qing Gao, Qi Wang.

**Methodology:** Qing Gao.

**Supervision:** Dafeng Lu.

**Visualization:** Qing Gao.

**Writing – original draft:** Qing Gao.

**Writing – review & editing:** Meng Wang, Bingdong Zhan.

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
