## [Decision Letter · Decision Letter 0]

26 Jul 2024

PONE-D-23-43491Fasting plasma glucose levels are associated with all-cause mortality and cancer death: a population-based retrospective cohort studyPLOS ONE

Dear Dr. Zhan,

Thank you for submitting your manuscript to PLOS ONE. After careful consideration, we feel that it has merit but does not fully meet PLOS ONE’s publication criteria as it currently stands. Therefore, we invite you to submit a revised version of the manuscript that addresses the points raised during the review process.  Please submit your revised manuscript by Sep 09 2024 11:59PM. If you will need more time than this to complete your revisions, please reply to this message or contact the journal office at plosone@plos.org. Please include the following items when submitting your revised manuscript:A rebuttal letter that responds to each point raised by the academic editor and reviewer(s). You should upload this letter as a separate file labeled 'Response to Reviewers'.A marked-up copy of your manuscript that highlights changes made to the original version. You should upload this as a separate file labeled 'Revised Manuscript with Track Changes'.An unmarked version of your revised paper without tracked changes. You should upload this as a separate file labeled 'Manuscript'.

We look forward to receiving your revised manuscript.

Kind regards,

Mahmoud M Werfalli, PhD

Academic Editor

PLOS ONE

Journal Requirements:

2. In the online submission form, you indicated that "The data sets generated and/or analysed during the current study are not publicly available due individual privacy information protection but are available from the corresponding author on reasonable request."

Reviewers' comments:

Reviewer's Responses to Questions

**Comments to the Author**

1. Is the manuscript technically sound, and do the data support the conclusions?

Reviewer #1: Yes

Reviewer #2: Yes

2. Has the statistical analysis been performed appropriately and rigorously? 

Reviewer #1: Yes

Reviewer #2: Yes

3. Have the authors made all data underlying the findings in their manuscript fully available?

Reviewer #1: Yes

Reviewer #2: No

4. Is the manuscript presented in an intelligible fashion and written in standard English?

Reviewer #1: Yes

Reviewer #2: No

5. Review Comments to the Author

Reviewer #1: This article is clearly structured, has some research significance, and the sample size is sufficient. The following are some problems that may need to be revised

1，The 2 in the unit of BMI in line 91 should be superscripted

2，Use a unified term for fasting plasma glucose and use abbreviations FPG, in line 152 and other lines

3，Please describe the data screening process in detail and how many people were excluded

4, Please conduct subgroup analysis by sex and age

Reviewer #2: #3 the data have not been made fully available, but the authors indicate "The data sets generated and/or analysed during the current study are not publicly available due individual privacy information protection but are available from the corresponding author on reasonable request." This seems reasonable to me.

#4 the article is in good shape, but would benefit from English language review. I have pointed out some issues below.

The authors state that “the requirement for informed consent was waived due to the retrospective design.” This does not make sense to me. If the data come from human subjects, whether prospective or retrospective, the individuals should have the opportunity to provide informed consent or to request that their data not be used.

Line 46 – what does this phrase refer to? “approximately 4.2 million adults aged 20 to 79 years [7].”

Lines 50-54 – in regard to the study from Korea, the study examined the relationship between glucose levels and mortality in individuals with prevalent diabetes, and it appears that the risks were highest in those with the highest glucose levels compared with 85–99 mg/dL (not compared with the lowest levels of glucose). [I did not check the results of the other papers cited.]

Lines 67-69 - did this study include all residents of Quzhou City in that time period? If not, how were the individuals selected?

Lines 71-73 need to be edited for clarity

Line 83-84 – what about former smokers who had quit less than six months ago, how are they categorized?

Line 90 – what were the “rigorous procedures” for collecting height and weight?

Line 104 – “lack of information on variables” is vague. What information was considered necessary in order for the individual not to be excluded?

Table 1 – I think the all-cause mortality and the cancer mortality numbers represent those who are still alive (I had assumed they represented those who had died, but the numbers don’t make sense that way.) Please clarify.

6. PLOS authors have the option to publish the peer review history of their article (what does this mean?). If published, this will include your full peer review and any attached files.

Reviewer #1: **Yes: **Kaizhi Bai

Reviewer #2: No

---

## [Author Response · Author response to Decision Letter 0]

16 Aug 2024

Reviewer 1

Thanks very much for taking your time to review this manuscript. We are very grateful to your generous comments for the manuscript. I will answer your comments one by one：

Response to comment:

This article is clearly structured, has some research significance, and the sample size is sufficient. The following are some problems that may need to be revised. 

1.The 2 in the unit of BMI in line 91 should be superscripted 

Thank you for your reminder, I have corrected this error.

2.Use a unified term for fasting plasma glucose and use abbreviations FPG, in line 152 and other lines

According to your suggestion, I checked the entire manuscript and standardized the terminology for fasting plasma glucose, using the abbreviation FPG throughout. I also checked the usage of other terms in the article and standardized "all-cause and cancer death" to "all-cause and cancer mortality."

3.Please describe the data screening process in detail and how many people were excluded 

I created a flowchart to illustrate the data screening process and the reasons and number of exclusions, making the analysis in the article clearer and more readable. 

4.Please conduct subgroup analysis by sex and age 

Thank you very much for your suggestion. I have added this part of the analysis, conducting subgroup analysis by age (< 60 years and ≥ 60 years) and gender to explore potential modifying effects. The subgroup analysis found that in different genders, the risk of all-cause death in the IFG and DM groups and the risk of cancer death in the DM group increased to varying degrees, but the risk of death in women was slightly higher than in men. The risk of hyperglycemia on all-cause mortality and cancer mortality was more pronounced in the <60 years age group. 

Reviewer 2

We would like to thank the reviewers from the bottom of our hearts for carefully reviewing and guiding the paper, which has led to a great improvement in the quality of the paper.

Response to comment:

#3 the data have not been made fully available, but the authors indicate "The data sets generated and/or analyzed during the current study are not publicly available due individual privacy information protection but are available from the corresponding author on reasonable request." This seems reasonable to me. 

#4 the article is in good shape, but would benefit from English language review. I have pointed out some issues below. 

1.The authors state that “the requirement for informed consent was waived due to the retrospective design.” This does not make sense to me. If the data come from human subjects, whether prospective or retrospective, the individuals should have the opportunity to provide informed consent or to request that their data not be used. 

This database comes from the 2015 resident health check-up data, which is part of a routine public health program. And in the retrospective analysis, we removed identifiable information, so the Quzhou CDC Ethics Review Committee waived the requirement for informed consent.

2.Line 46 – what does this phrase refer to? “approximately 4.2 million adults aged 20 to 79 years [7].”

This sentence refers to the number of adults aged 20 to 79 who died from diabetes in 2019. I have revised this sentence to make the content clearer

3.Lines 50-54 – in regard to the study from Korea, the study examined the relationship between glucose levels and mortality in individuals with prevalent diabetes, and it appears that the risks were highest in those with the highest glucose levels compared with 85–99 mg/dL (not compared with the lowest levels of glucose). [I did not check the results of the other papers cited.] 

Thank you for your help. We apologize for our carelessness. Upon careful reading, it was found that in this prospective study in Korea, the relationship between fasting blood glucose and mortality in diabetic patients was examined. It was found that compared with 85-99 mg/dL, the multivariable adjusted risk ratio for hypoglycemia (<65 mg/dL) was 1.46. Avoiding hypoglycemia can improve the survival rate of diabetic patients. So we have revised it to "A prospective cohort study in Korea on the relationship between diabetic patients and mortality found that fasting blood glucose and all-cause mortality showed a U-shaped relationship, and hypoglycemia could increase the mortality rate of diabetic patients." 

4.Lines 67-69 - did this study include all residents of Quzhou City in that time period? If not, how were the individuals selected? 

Thank you for your question. This study did not include all residents of Quzhou City during that time period. Instead, it included individuals who voluntarily participated in the 2015 health check-ups organized by the Quzhou City Health Bureau. These health check-ups were part of a routine public health initiative, and participation was open to all residents. Participants were not randomly selected; however, the program was widely publicized and accessible to all segments of the population, which helps to ensure a diverse and representative sample of the community. Given the large sample size and the inclusivity of the health check-up program, we believe the cohort provides a reasonably representative cross-section of the Quzhou City population.

5.Lines 71-73 need to be edited for clarity 

We have re-edited this content, and we want to express that the information in this database was entered into the Resident Health Check Information Database after quality control of the original data by trained professionals. This makes it easier for readers to understand the content.

6.Line 83-84 – what about former smokers who had quit less than six months ago, how are they categorized? 

We categorized smoking status into three groups: never smoked, former smoker, and current smoker. Individuals who had previously smoked but quit for more than six months were classified as former smokers. If they had quit for less than six months, they were judged on the basis of their past smoking habits—if they had smoked at least 100 cigarettes, they were considered current smokers; otherwise, they were classified as never smoked. To avoid ambiguity, I changed 'current smoking' to 'smoking' for clearer expression. 

7.Line 90 – what were the “rigorous procedures” for collecting height and weight? 

We replaced the phrase "rigorous procedures" with specific content, referring to the fact that the height and weight of all participants were measured using standard instruments while they were wearing light clothing and had removed their shoes and hats.

8.Line 104 – “lack of information on variables” is vague. What information was considered necessary in order for the individual not to be excluded? 

The lack of variable information refers to the lack of records on smoking, drinking, and physical exercise. In the article, we have clarified this to avoid ambiguity and confusion for readers.

9.Table 1 – I think the all-cause mortality and the cancer mortality numbers represent those who are still alive (I had assumed they represented those who had died, but the numbers don’t make sense that way.) Please clarify. 

We apologize for the mistake. All-cause mortality and cancer mortality represent those who have died. We have corrected this part.

---

## [Editor Report · Decision Letter 1]

16 Sep 2024

Fasting plasma glucose levels are associated with all-cause and cancer mortality: a population-based retrospective cohort study

PONE-D-23-43491R1

Dear Dr. Bingdong Zhan

We’re pleased to inform you that your manuscript has been judged scientifically suitable for publication and will be formally accepted for publication once it meets all outstanding technical requirements.

Kind regards,

Mahmoud M Werfalli, PhD

Academic Editor

PLOS ONE

---

## [Editor Report · Acceptance letter]

18 Sep 2024

PONE-D-23-43491R1 

PLOS ONE

Dear Dr. Zhan, 

I'm pleased to inform you that your manuscript has been deemed suitable for publication in PLOS ONE. Congratulations! Your manuscript is now being handed over to our production team.

Kind regards, 

on behalf of

Dr. Mahmoud M Werfalli 

Academic Editor

PLOS ONE